# Atlantic sediments reveal interacting environmental and physiological controls on coccolithophore calcite production

Alba González-Lanchas [1] ✉, Karl-Heinz Baumann [2], Heather. M. Stoll [3], José-Abel Flores [4], Miguel. A. Fuertes [5] & Rosalind. E. M. Rickaby [1]

Coccolithophores contribute 20-80% of the open ocean's total calcite production, playing a pivotal role in the marine carbon cycle. Links between environment, coccolithophore physiology and calcite production remain unclear due to challenges in extrapolating culture experiments to sedimentary nannofossil records. Here, we develop a framework to reconstruct physiology and calcite production of dominant coccolithophore species from sedimentary records. Using well-preserved Atlantic surface sediments, this study establishes factors controlling coccolithophore calcite production via measurements of species composition, primary production, growth (µ), and calcification rates. Contrasting µ-calcification relationships of major groups indicate differing carbon requirements. Optimal µ is the primary control on group-specific maximum calcite production, defining a meridional bimodal structure with a boundary at ~40°N, aligned with oceanic physicochemical gradients. Group-specific cellular carbon demand relative to supply show that this boundary separates reaction-limited cells to the north from mass-transport-limited cells to the south, and likely migrates latitudinally with changing ocean carbon.

Carbon partitioning between surface and geologic reservoirs is essential for regulating atmospheric $CO_2$ concentrations, which directly impact Earth's climate over long timescales[1]. Marine pelagic calcifiers use dissolved inorganic carbon (DIC) in conjunction with $Ca^{2+}$ ions to form calcium carbonate ($CaCO_3$) biominerals, primarily in the form of calcite. Sinking, deposition and accumulation of pelagic calcite on the ocean floor create a pathway for long-term carbon alkalinity storage. Consequently, marine pelagic production marks the initial stage in a sequence that includes the deposition and dissolution of calcite, playing a key role in the long-term operation of both the marine and global carbon cycles[2,3].

Among pelagic calcifiers, coccolithophores contribute 20-80% of marine biogenic calcite[4–6]. This phytoplankton group incorporates carbon into both calcification and photosynthesis, producing particulate inorganic and organic carbon (i.e., PIC and POC). These dual metabolic pathways underscore their integral role in the biogeochemical cycling of carbon within the ocean surface[7,8]. Coccolithophore calcification (the intracellular nucleation of calcite biominerals) materializes the production of minute platelets called "coccoliths" that are later extruded outside the cell to form an interlocking, exoskeletal coccosphere[9]. The high density of calcite and sinking coccoliths along the water column drives a net export of PIC and POC to the deep ocean[10,11]. The cellular PIC/POC ratio, or the population-level "rain ratio", both interact with the surface seawater chemistry and modify deep-ocean chemistry by influencing the rates of accumulation and burial of carbon in sediments. Variability in coccolithophore PIC/POC can alter marine carbonate compensation and dissolution patterns[12–14], affecting the carbon cycle on geological time

[1]Department of Earth Sciences, University of Oxford, Oxford, UK. [2]Department of Geosciences, University of Bremen, Bremen, Germany. [3]Department of Earth and Planetary Sciences, ETH Zürich, Zürich, Switzerland. [4]Department of Geology, University of Salamanca, Salamanca, Spain. [5]Department of Didactics of Mathematics and Experimental Sciences, University of Salamanca, Salamanca, Spain. ✉e-mail: alba.gonzalez-lanchas@earth.ox.ac.uk

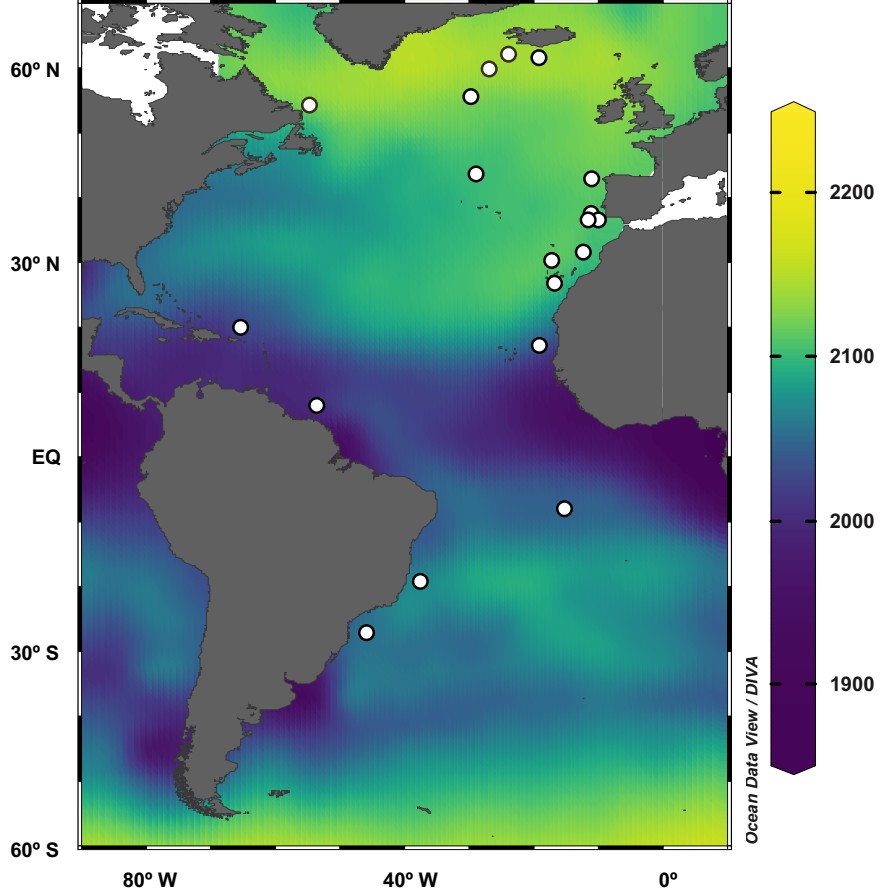

**Fig. 1 | Surface sediment locations and spatial variability of surface carbon.** Spatial distribution of surface sediment samples in this study (white dots) overlaid on a map of total dissolved inorganic carbon in surface seawater (i.e., TCO₂; μmol/kg) for January extracted from the gridded datafile by Takahashi et al.[104], in which the values were calculated from total alkalinity (TA) and surface $p$CO$_2$. The TCO$_2$ values are represented and visualized here using Ocean Data View[95].

scales[15]. Given the pivotal role of coccolithophores in the marine and global carbon cycle[16] and climate feedbacks[17,18], understanding the interaction between natural environmental drivers and physiological processes that regulate their calcite production constitutes a fundamental priority in current scientific research.

The fixation of carbon by coccolithophores is primarily fuelled by photosynthesis. Calcite formation is intricately regulated by physiological mechanisms and simultaneously modulated by external environmental factors[19,20]. Although numerous laboratory studies have examined how various conditions influence physiology and calcite production[7,8,21–23], the principal drivers governing these processes in the natural environment remain incompletely resolved (see Brownlee et al.[24] and references therein). In particular, the dynamic coupling between PIC and POC production (the relationship between photosynthesis and calcification) and its response to seawater carbon availability and environmental physicochemical stressors is not fully understood. From a geological perspective, temperature, light, nutrients, and seawater carbon chemistry have all been suggested to play an important role[14,25–30].

Culture experiments manipulate single strains with uniform genomes over short periods, preventing genetic exchange or phenotypic plasticity. While these studies provide valuable insights into physiology, their phenotypic homogeneity limits extrapolation to the natural ocean and geologic time scales, over which adaptation and selective pressures drive long-term changes in both individuals and communities (see McClelland et al.[31] and references therein). Coccolithophore calcite production depends on net population density and species composition (i.e., population dynamics), as well as on individual growth (μ) and cellular calcification rates[11,27,32]. With these factors differing among coccolithophore species and groups, focused research is needed to resolve their integrated control on coccolithophore calcite production under natural conditions. From the characterization of assemblage composition and group-specific absolute abundances, micropaleontology of coccoliths preserved in sediments provides a route for the assessment of population dynamics and primary productivity changes (see Bolton and Stoll[33] and references therein). Likewise, coccolith morphometry, namely size and allometry-normalised thickness change, provides insights into cell dimensions[34], μ[35,36] and calcification intensity[30,31,37,38].

In order to understand the environmental and physiological controls on coccolithophore calcite production across the Atlantic Ocean, we have generated micropaleontological and morphometric analysis of the well-preserved coccolith assemblages in surface sediment samples from a meridional transect (Fig. 1 and Supplementary Figs. 1 and 2). This study examines the coccolithophore species/groups small *Gephyrocapsa*, large *Gephyrocapsa, Helicosphaera* spp., *Calcidiscus* spp., and *Coccolithus pelagicus*, which are the dominant taxa in modern to Holocene coccolithophore communities (Supplementary Notes 3 and references therein). Following a well-established culture-based physiological criteria (Methods) coccolithophores are categorized by relative rates of PIC to POC production, as the low PIC/POC group (*Gephyrocapsa* spp.) and the high PIC/POC group

(*Helicosphaera* spp., *Calcidiscus* spp., and *C. pelagicus*). Combining micropaleontological census data of coccolith assemblages with morphometric analyses on individual coccoliths, we characterize group-specific primary productivity, μ, and calcification intensities. Their respective contribution to coccolithophore calcite production comprises the numerical abundance combined with the group-specific calcite per coccolith (Methods).

Here, we show contrasting μ-calcification relationships among major coccolithophore groups in the Atlantic Ocean, indicative of differing carbon requirements that determine a gradient between reaction-limited calcification, in the higher latitudes, and mass-transport-limited calcification, in the lower latitudes. These regimes determine the biogeographic distribution of the dominant groups and a bimodal structure of their dominance on coccolithophore calcite contribution to sediments, with a boundary at ~40°N. This bimodality is driven by the group-specific optimal μ within each group and aligns with large-scale oceanic physicochemical gradients. A contrasting physiological-environmental correspondence between groups likely reflects the ratio between metabolic rate (cell demand) vs. carbon substrate (cell supply) modes in the low and high PIC/POC specimens, respectively. Having characterized these relationships from surface sediments, with the use of micropaleontological and morphometric techniques, this study introduces a framework for interpreting nannofossil sedimentary records in terms of carbon utilisation in studies of past geological intervals.

## Results and Discussion
### Group-specific growth rate (μ) and calcification intensity
Multiple lines of evidence indicate good preservation and representativeness of the sedimentary coccolith assemblages in this study of natural Atlantic coccolithophore community structure and surface production patterns (Methods and Supplementary Notes 1 and 3). The total absolute abundance of coccoliths in sediments (N; coccolith g$^{-1}$; Methods) appears to be a good proxy for coccolithophore productivity across the studied transect. These measurements show a peak in the Atlantic mid-latitude to subtropical regions (50°–20°N and south of 20°S; Fig. 2a), a pattern that aligns with satellite- and model-based estimates of annual primary productivity in the modern Atlantic (Supplementary Notes 3 and references therein). The spatial correspondence, combined with the minimal impact of dissolution and post-depositional preservation effects, supports the use of N as a qualitative proxy for variations in surface ocean coccolithophore productivity (Supplementary Notes 1 and 3). Having demonstrated the minimal impact of dissolution, morphometric profiles and indexes, such as Size Normalized Thickness (SN Thickness), are shown to be a reliable indicator of calcification intensity (Supplementary Notes 1, 2 and Supplementary Table 3). The SN Thickness measurements here correspond with other derivations of calcification, such as the elliptical shape factor (kse) and a morphometric PIC/POC calculation for the low PIC/POC group (Supplementary Methods 2 and Supplementary Fig. 6).

The low PIC/POC group is the most abundant across latitudes (Fig. 2a, e, f; Supplementary Fig. 7), a situation of dominance typical of modern to Holocene Atlantic coccolithophore communities (Supplementary Notes 3 and references therein). This reflects both the broad ecological tolerance of this group as well as the good preservation and representativeness of our sedimentary records. From an independent derivation of μ (μ$_{size}$), based on the inverse relationship between cell-size and μ[35,36] (Methods), both small and large *Gephyrocapsa* display corresponding trends in productivity (N) and growth rate (μ$_{size}$) across latitudes. Maxima values are seen for both parameters in the Atlantic mid-latitudes and subtropics (Fig. 3a, d, f, i). These two independent proxies, N and μ$_{size}$, appear to be interchangeable as qualitative indicators of *Gephyrocapsa* μ[39]. For the low PIC/POC group, estimates of calcification intensity, from

allometry-normalised thickness change (Methods, Supplementary Methods 2 and Notes 2), parallel μ across the latitudes (Fig. 3a–c, f–h). In these relatively small-sized cells, carbon availability appears to be sufficient for calcification to keep pace with faster μ. This coupling between intensified calcification and faster μ indicates a balanced flow of energy and carbon within cellular processes[20,40,41], ultimately reflecting conditions of elevated and efficient utilisation of carbon in those regions (Fig. 3).

The relationship between faster μ and intensified calcification of the low PIC/POC group in the Atlantic mid-latitudes and subtropics aligns with the area of maximum calcification derived from satellite observations and the calcite production potential of *G. huxleyi* (i.e., species included within the small *Gephyrocapsa* group)[42]. A similar pattern emerges in the fossil record. During the mid-Brunhes interval of the late Pleistocene (~400,000 years ago), *Gephyrocapsa* specimens exhibited significantly intensified calcification, as indicated morphometrically by allometry-normalised thickness increase (SN Thickness and morphometric PIC/POC; in Gonzalez-Lanchas et al.[14]). This intensified calcification coincides with elevated μ, from enhanced N[14], and is further supported by independent geochemical proxies of coccolithophore μ, as the Sr/Ca and δ$^{13}$C measured in coccolith calcite[13]. Together, these lines of evidence suggest a link between faster μ and intensified calcification in the low PIC/POC group, common to extant and extinct phylogenetically related specimens[43], from the late Pleistocene up to the present. This group's overall lower carbon requirements, potentially due to its smaller cell dimensions (Supplementary Figs. 4 and 5) and/or its higher capacity for carbon acquisition and utilisation[40] enable it to more efficiently allocate carbon between cellular processes. As a result, environmental carbon availability seldom constrains cellular demand.

Enhanced presence of the high PIC/POC group is observed toward the Atlantic upper-mid to high latitudes in this study (Fig. 2a-d, f; Supplementary Fig. 7), in full agreement with the biogeographic distribution of those groups in modern to Holocene Atlantic coccolithophore communities (Supplementary Note 3 and references therein). Higher productivity of the high PIC/POC group (N), likely linked with faster μ, is observed at the upper mid to high latitudes above ~40°N (Fig. 2a-d). In this group, morphometric analyses are not yet well established for calcification intensity. Above ~40°N, the highest N of this group are coincident with their smaller cell size (inferred from coccolith size reduction; Supplementary Fig. 5), consistent with the well-established inverse relationship between cell size and μ[35]. In the high PIC/POC producers, a higher N is related to reduced SN Thickness (most significantly in *C. pelagicus* and *Calcidiscus* spp.; Fig. 2b, c and Supplementary Fig. 9). In contrast to the low PIC/POC group, the highest μ appear to induce reduced calcification intensity (Fig. 2b–d and Supplementary Fig. 9). Higher carbon requirements in high PIC/POC specimens[44], driven in part by their relatively larger cell dimensions and a more passive style for carbon acquisition, likely underpins their distinctive carbon physiology. Faster μ of this group may outpace the supply of carbon, leading to a reduced calcification cell quota[45]. This is well expressed above ~40°N as an allometric normalized thickness reduction (SN Thickness reduction; Fig. 2b–d). Conversely, slower μ of the high PIC/POC group, in regions south of ~40°N, may allow carbon supply to remain within the limits of cell demand, allowing enhanced calcification toward those regions (Fig. 2b–d).

In summary, micropaleontology and morphometry of well-preserved coccolith assemblages from Atlantic surface sediments reveal that increased productivity and faster μ in low PIC/POC specimens are accompanied by efficient carbon utilisation and allocation, and higher calcification intensity (Fig. 2e, f and Fig. 3a–c, f–h, k, m). The opposite pattern is observed in high PIC/POC species (Fig. 2b-d and Supplementary Fig. 9). These contrasting trends between μ and calcification intensity reflect differing carbon demand

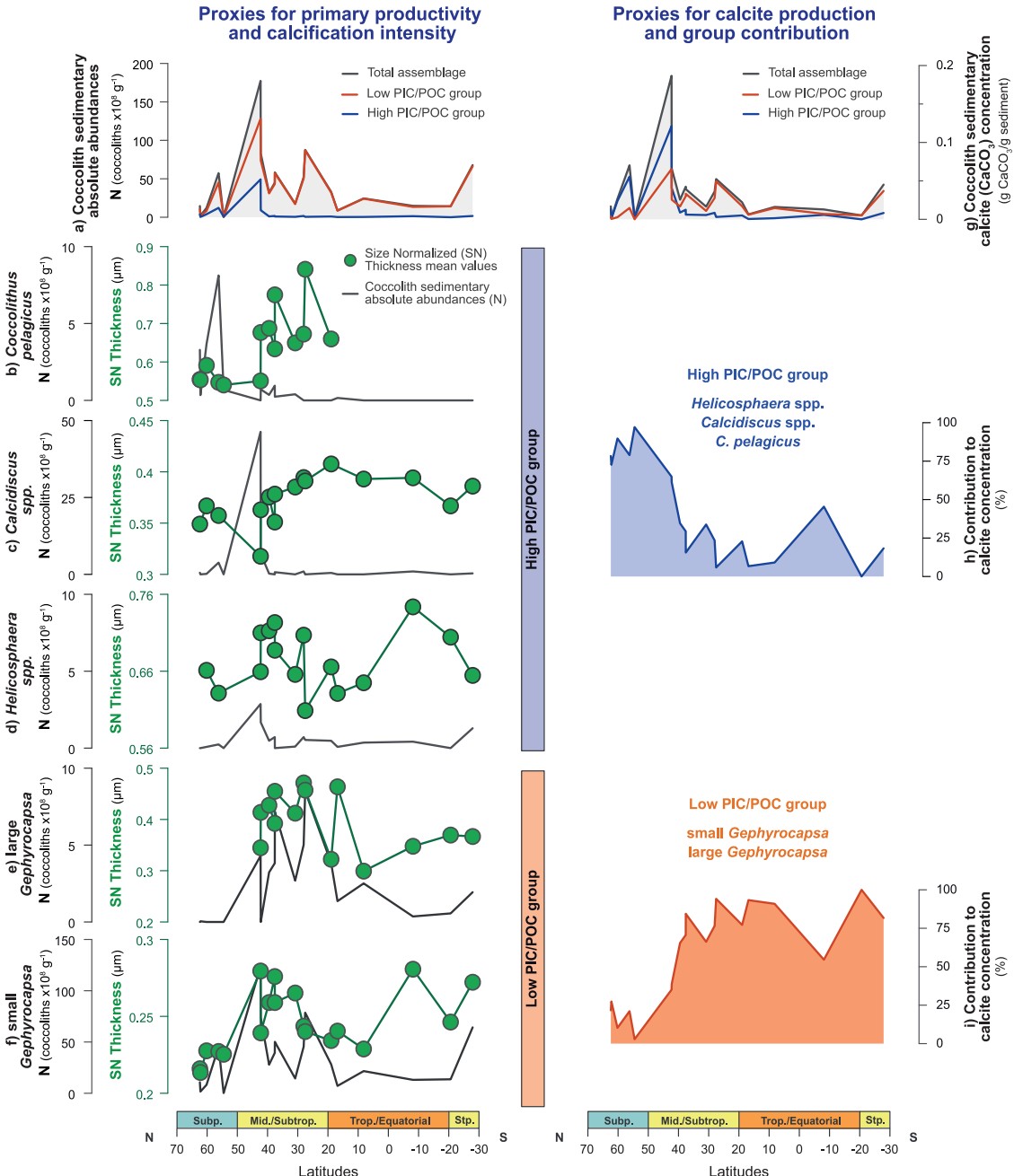

**Fig. 2 | Latitudinal variability in species and group-specific micro-paleontological and morphometric proxies. a** Primary productivity estimate from the absolute abundance of coccoliths in sediments (N; coccolith x $10^8$ g⁻¹; Methods), corresponding to the total coccolithophore assemblage (grey), the low PIC/POC group (orange), and the high PIC/POC group (blue). Aligned primary productivity (N) and calcification intensity (SN Thickness; μm) profiles of the taxa analysed in this study: **b** *C. pelagicus*; **c** *Calcidiscus* spp.; **d** *Helicosphaera* spp.;

**e** large *Gephyrocapsa* and **f** small *Gephyrocapsa*. **g** Coccolithophore calcite production estimate, from coccolithophore calcite concentration in sediments (pg $CaCO_3$ g⁻¹ sediment; Methods), corresponding to the total assemblage (grey), the low PIC/POC group (orange), and the high PIC/POC group (blue). Group-specific relative contribution to coccolithophore calcite production (%) of **h** high PIC/POC group (blue) and **i** low PIC/POC group (orange).

and intracellular carbon allocation strategies between the low and high PIC/POC groups, which shape their distinct biogeographic distributions. To extend the implications of these contrasting physiological modes to their impact on ocean biogeochemistry, we next explore group-specific contributions to total coccolithophore calcite production.

**Bimodal structure of coccolithophore calcite production**

The absolute abundance (N) of the low PIC/POC in sediments consistently surpasses that of the high PIC/POC group at all locations

(Fig. 2a and Supplementary Notes 3). Physiologically, this reflects the faster μ of the small-sized cells of the low PIC/POC specimens[35,46,47] and agrees with the characteristics of the biogeographic distribution of groups in the modern ocean and in Holocene Atlantic sedimentary records (Supplementary Notes 3 and references therein). Nonetheless, we found a bimodal distribution in the dominant contributions of the low and high PIC/POC groups to coccolithophore calcite production across latitudes (Fig. 2g–i). The high PIC/POC group, primarily driven by *C. pelagicus* and *Calcidiscus* spp. taxa, dominates coccolithophore calcite production at latitudes above ~40°N, characterized by their

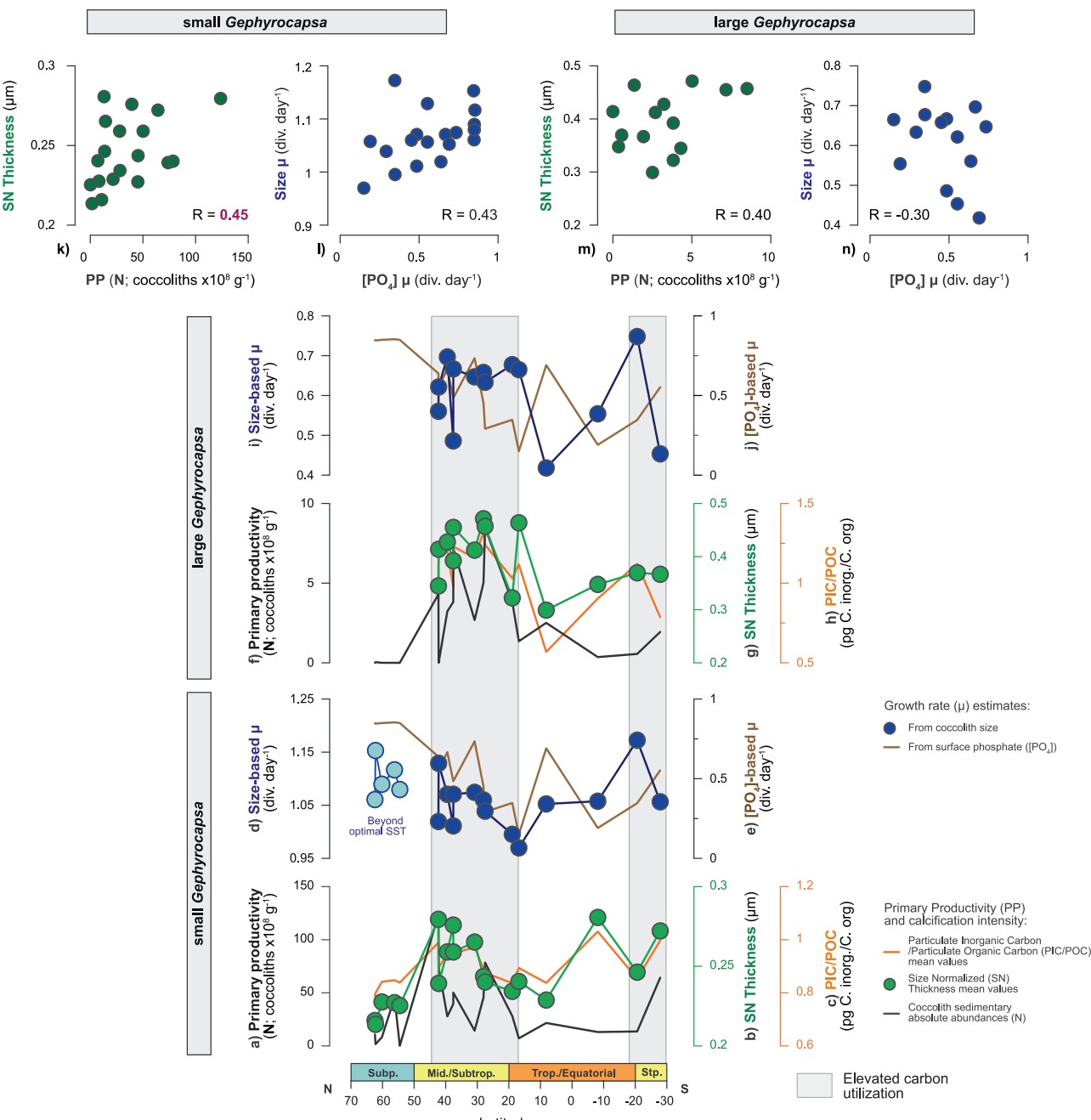

**Fig. 3 | Latitudinal relationships between growth rate (μ) and calcification intensity estimates for the taxa composing the low PIC/POC group. a** Primary productivity of small *Gephyrocapsa* (N; coccolith x $10^8$ g$^{-1}$). Calcification intensity of small *Gephyrocapsa* from profiles **b** Size Normalized (SN) Thickness (μm) and **c** morphometric Particulate Inorganic to Organic Carbon (PIC/POC; pg C inorganic / C organic); **d** small *Gephyrocapsa* size-based μ estimate (μ$_{size}$; div. day$^{-1}$), compared with **e** independent μ parametrization from surface [PO$_4$] (μ$_{[PO4]}$; div. day$^{-1}$); **f** Primary productivity of large *Gephyrocapsa* (from N; coccolith x $10^8$ g$^{-1}$). Calcification intensity of large *Gephyrocapsa* from profiles **g** Size Normalized (SN) Thickness (μm) and **h** morphometric Particulate Inorganic to Organic Carbon (PIC/POC; pg C inorganic/C organic); **i** large *Gephyrocapsa* size-based μ estimate (μ$_{size}$;

div. day$^{-1}$), compared with **j** independent μ parametrization from surface [PO$_4$] (μ$_{[PO4]}$; div. day$^{-1}$). Scatter plots showing the relationship between **k** primary productivity (i.e., PP, from N) and calcification intensity (SN Thickness) of small *Gephyrocapsa*; **l** μ $_{size}$ small *Gephyrocapsa* and μ $_{[PO4]}$; **m** primary productivity (PP) and calcification intensity (SN Thickness) of large *Gephyrocapsa*; **n** μ $_{size}$ large *Gephyrocapsa* and μ $_{[PO4]}$. The size-based μ and [PO$_4$]-based μ parametrizations are independent and calculated with the regressions by Zhang et al.[36] and Krumhardt et al.[60], respectively (see Methods). Independent μ parametrizations based on environmental parameters SST and [CO$_2$] are included in the Supplementary Fig. 8. See Methods for information about environmental data, corrections, and calculations.

faster (or optimal) μ, despite their reduced calcification intensity (Figs. 2b–d; g, h). In turn, the low PIC/POC group prevails as the dominant contributor at latitudes south of ~40°N, that encompass the Atlantic mid-latitudes and subtropical regions of faster (optimal) μ and intensified calcification of these specimens (Fig. 2e, f, g, l; Fig. 3). In the Atlantic ocean, there appears to be a boundary ~40°N in the

contributions to calcite production by coccolithophores between these two dominant populations (Fig. 2 and Supplementary Fig. 10).

Coccolithophore calcite production is estimated by considering the taxa-specific coccolith abundances (N) and the taxa-specific average coccolith mass at each location (see Methods). Variations in these two factors are dynamic elements that lead to changes in calcite

production and the relative contribution of groups to coccolithophore CaCO₃. The low PIC/POC group has lower average coccolith size and mass values and a narrower range of variability in values between samples (Supplementary Figs. 4, 5). The higher N of the low PIC/POC specimens, which reflects faster μ, is the main factor influencing their greater contribution to coccolithophore calcite production (Fig. 2e, f, g, i). While the high PIC/POC group shows greater average size and mass values, and a wider range of variability between samples (Supplementary Figs. 4 and 5), its higher contribution to net coccolithophore calcite production appears to be, likewise, associated with enhanced N, indicative of faster μ (Fig. 2b–d, g, h). In sum, optimal μ is the primary driver of maximum group contribution to the net coccolithophore calcite production for both the low and high PIC/POC groups in natural conditions (Fig. 2). The higher calcite mass per coccolith of the high PIC/POC specimens allows this group to make a more significant contribution than the low PIC/POC specimens under similar (or lower) μ, as seen between ~40 and 50°N, which exhibit the highest coccolithophore calcite contribution to sediments (Fig. 2g). The high PIC/POC specimens play a key role in calcite contribution within mixed coccolithophore communities in those North Atlantic upper-mid-latitudinal settings. This observation provides fidelity and field support for the connection between culture experiments and projections made by Daniels et al.[46], highlighting the significance of natural coccolithophore diversity in driving increased oceanic calcite production.

The bimodality between high PIC/POC cells to the north and low PIC/POC cells to the south, separated by a boundary at ~40 °N (Fig. 2g and Supplementary Fig. 10), combined with the discovery that coccolith calcite crystal growth may be transport limited[48], hint at the cellular controls underlying the PIC/POC ratio, and even the controls on the morphology of coccoliths. By analogy with snowflakes, the more dendritic style of calcite formation in the low PIC/POC group suggests mass transport limitation of their calcification, compared to the reaction limitation of the more space-filling platey coccoliths produced by the high PIC/POC species[49]. Extending the experimental inverse relationship between μ and size in coccolithophores[35] to the marine environment, points to large and slow growing cells north ~40 °N and small and faster dividing cells to the south of this boundary (Fig. 2g–I and Supplementary Fig. 10). Assuming that metabolic rate is proportional to μ, higher metabolic rates in the faster growing low PIC/POC specimens are likely expressed at the cellular level, in a faster transit time of the coccolith vesicle across the cell, particularly when the cells are small. Faster metabolism and calcification rates in low PIC/POC specimens entail a higher demand for carbon substrate. As a result, the time available for the accumulation of the substrates for calcification is diminished. Even though the cells of the low PIC/POC group are smaller (Supplementary Figs. 4 and 5), with higher surface area and substrate transport rates, the per coccolith rate of calcification is, nonetheless, limited by the mass transport of the substrate into the cell; this results in the dendritic style of crystal formation in this group[49]. By contrast, in the high PIC/POC group, with comparatively slower μ and metabolic rates, the slower turn-over of the coccolith vesicle allows sufficient time for the substrate transport across the cell to accumulate to form platey style crystals[49], such that the precipitation reaction alone limits the calcification rate, as long as there is sufficient carbon in the environment to meet the demand. These dynamics of carbon within species of different sizes and morphometries, extracted from the μ-calcification relationships and bimodality of CaCO₃ contribution in this research, are well captured in their characteristic isotopic vital effects[44,50]. To a first order, cell size is proportional to the degree of space-filling crystallisation of each coccolith, which is mechanistically linked to the residence time of carbon within the calcifying vesicle, a function of metabolic rate vs. carbon substrate. The distinction between mass transport limitation of cellular calcification, south ~40 °N, and reaction limitation, north of this boundary

(Fig. 2g–i), likely reflects the size-dependent and environmentally controlled metabolic rate vs. carbon substrate ratio in the low and high PIC/POC groups.

## Environmental controls on coccolithophore group-specific calcite production

The bimodal structure of coccolithophore calcite production across the Atlantic Ocean meridional transect, which reveals the contrasting metabolic rates vs. carbon substrate ratio of the low and high PIC/POC groups, fully aligns with the classic assessments about the biogeographic distribution of dominant coccolithophore groups in the Atlantic Ocean, from the Holocene into the present day (Supplementary Notes 3 and references therein). Our study offers an additional perspective on this bimodality, as indicative of a gradient between reaction-limited growth of cellular calcite in the higher latitudes and transport-limited growth across the lower latitudes. This dictates the latitudinal position of a boundary in group-specific dominance over the coccolithophore calcite contribution to sediments (Fig. 2g–i).

Environmental conditions shape the physiological performance and biogeographic distribution of groups, yet the activity of these groups may, in turn, modulate local surface seawater chemistry. Natural ocean physicochemical parameters are closely linked to the function of the biological carbon pump and exhibit strong natural covariation in the surface ocean due to the relative importance of mixing versus productivity[51–54]. We observe the bimodal structure is consistent with the correlated parameters across the latitudinal transect of the ratio of Dissolved Inorganic Carbon to Total Alkalinity ratio (DIC/TA), Sea Surface Temperatures (SST), the carbon isotope ratio of the surface seawater DIC (δ¹³C_DIC) and surface [PO₄] (Fig. 4 and Supplementary Fig. 10). In order to evaluate how the environment and coccolithophore groups co-interact, we will next explore how the variability in those surface ocean physicochemical parameters influence the reconstructed optimal μ and calcification intensities of groups on either side of this boundary (i.e. within the conditions conducive to the high versus low PIC/POC groups).

For the low PIC/POC group, optimal μ, intensified calcification, and a greater contribution to calcite production is recorded in the mid-latitudes and subtropical Atlantic environments. The surface [CO₂] and [HCO₃⁻], the main substrates for photosynthesis and calcification[55], appear relatively reduced at these environments, while TA is relatively high (integrated as relatively reduced DIC/TA ratio; Fig. 5g, h, j). The low PIC/POC species appear to compete more effectively at lower carbon availability, a consideration that builds on experimental observations of a higher affinity of low PIC/POC specimens for the utilisation of HCO₃⁻ as a substrate[56,57]. Non discriminatorily, the rapid μ, but lower calcification demand of this group, could partially contribute to the [HCO₃⁻] depletion relative to alkalinity in the water column in those environments.

The low PIC/POC species appear to be better adapted to regions that are transitional between the well-mixed high latitudes and the stratified lower latitudes (Fig. 5d). In those sectors, irradiance levels are intermediate between the limited photoperiods in high latitudes and the excessive irradiance exposure in low latitude settings. In the regions of optimal μ, intensified calcification and a greater contribution of the low PIC/POC group, the SST values are temperate, ranging between 10 to 25° (Fig. 5c). We acknowledge the limitations of assigning absolute environmental values to the sedimentary trends, but the correspondence of this absolute SST range with the optimal temperature range for the μ of *Gephyrocapsa* coccolithophores in culture experiments provides confidence in these relationships[58,59]. Importantly, our determination of μ_size for the small *Gephyrocapsa* specimens corresponds with the trend in the independently inferred μ, based on surface phosphate concentrations[60] (Fig. 3d, e, and I and Supplementary Table 5). As such, surface nutrient ([PO₄]) stands as a control on the small *Gephyrocapsa* specimens, except at the high

latitudes beyond the optimal temperature range for this group (Fig. 3d, e). Higher affinity of small *Gephyrocapsa* for relatively nutrient-depleted surface environments, across the mid-latitudes (Fig. 5e), may reflect their advantages for nutrient uptake. In contrast, the large *Gephyrocapsa* group shows a weaker connection with the surface [PO$_4$] and also with the [CO$_2$] and SST, when μ$_{size}$ is independently inferred for this group (Fig. 3i, j, n and Supplementary Table 5). This overall lower dependence on surface environmental parameters likely results from a wider range of vertical habitat depth of species such as *G. oceanica*[61–64] (i.e., the dominant species within the large *Gephyrocapsa* group). Higher light requirements of the small versus large *Gephyrocapsa* groups[65,66] can explain their differences in vertical habitat depths and a higher affinity of the small *Gephyrocapsa* specimens to the surface-most parameters. Together with this, a broader ecological niche of the large *Gephyrocapsa* (as for the modern *G. oceanica*)[26] is possibly conditioned by the higher intraspecific diversity within this taxa[67].

Other parameters, such as surface salinity (i.e, SSS), show latitudinal correspondence with μ and calcification intensity in the low PIC/POC species (Fig. 5i), suggesting a complementary role for both the small and large *Gephyrocapsa* groups. Primarily governed by physical processes (i.e., precipitation/evaporation and riverine input), both SSS and TA parameters are highly co-variant in the surface ocean (Supplementary Fig. 11)[68]. The concentrations of cations comprising SSS, including Ca$^{2+}$, Na$^+$, K$^+$, and Mg$^{2+}$, vary conservatively with temperature and pressure[69,70], allowing the consideration of SSS as a qualitative estimate of net cation concentration across latitudes (Fig. 5i). The low and stable [Ca$^{2+}$] in the modern ocean[71] suggests its variability alone is unlikely to be a primary limiting factor controlling calcification. However, we suggest the enhanced net cation concentration in the Atlantic mid-latitudes and subtropics (Fig. 5i) can positively influence the metabolism of the low PIC/POC group, by facilitating cellular transport of Ca$^{2+}$ and activating enzymes involved in both photosynthesis and calcification functions[19,72,73]. A positive correlation between calcification intensities, μ, and carbon utilisation of small and large *Gephyrocapsa* is observed with δ$^{13}$C$_{DIC}$ and SSS, respectively (Fig. 5k, l). Small *Gephyrocapsa* specimens appear more responsive to nutrient availability (i.e., slightly elevated surface [PO$_4$] at the mid latitudinal to subtropical regions) and/or to the efficiency of nutrient utilisation, while the large *Gephyrocapsa* group appears to be more responsive to cation availability. We suggest this may be related to differences in cell dimensions of these two subgroups and/or their slightly different metabolic demands.

For the high PIC/POC group, optimal μ and a greater contribution to calcite production are observed at latitudes above ~40°N (Fig. 2g, h). The surface seawater [CO$_2$] and [HCO$_3^-$] appear relatively higher in those regions, while TA is relatively low (i.e., integrated as a relatively reduced DIC/TA ratio; Fig. 6i–k). A stronger link between the high PIC/POC group and natural conditions of higher seawater CO$_2$ availability is suggested, as supported by both the correspondence in this research and the elevated CO$_2$ requirement observed within this group under experimental conditions[44].

The regions for optimal μ and a greater contribution of the high PIC/POC group to calcite production, above the ~40°N latitudes, are characterized by vigorous vertical mixing, cooler SST values, below 15 °C, and a shorter and highly variable photoperiod toward the high latitudes (Fig. 6e, f). Evidence of reduced nutrient utilisation relative to supply is indicated by the elevated [PO$_4$], together with relatively light δ$^{13}$C$_{DIC}$ in those environments (Fig. 6g, h). The N profiles, indicative of μ for the three varieties within the high PIC/POC group, *C. pelagicus*, *Calcidiscus* spp. and *Helicosphaera* spp., reveal a north-to-south gradient in optimal values along the transect (Fig. 2b–d). Higher N *C. pelagicus* is found in the coldest SSTs of the high-latitudes, with enhanced N *Calcidiscus* and *Helicosphaera* spp. toward milder SSTs in the upper mid-latitudes (Fig. 6e). This latitudinal trend fully reflects the

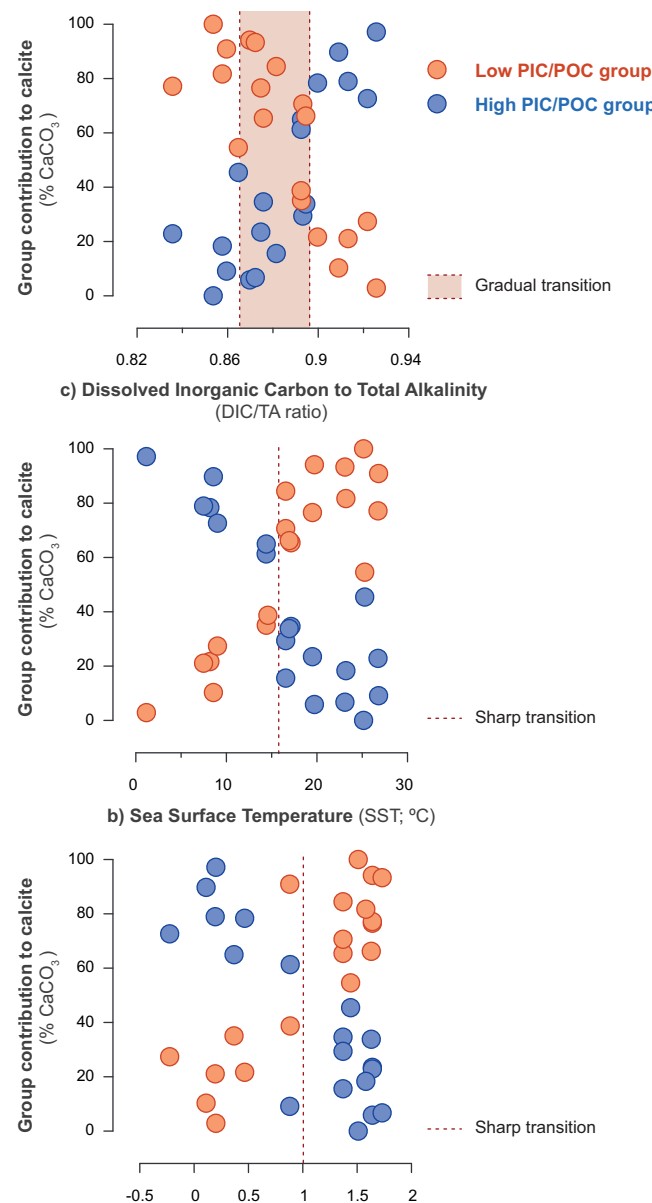

**Fig. 4 | Correspondence between the relative contribution of groups to coccolithophore calcite production and independent environmental parameters.** **a** Seawater carbon isotopic ratio of DIC (δ$^{13}$C$_{DIC}$; ‰); **b** Sea surface temperatures (SST; °C); **c** Ratio of Dissolved Inorganic Carbon to Total Alkalinity (DIC/TA ratio). The relative contribution of the low and high PIC/POC groups is represented with orange and blue dots, respectively. Note that the proportions of the two groups are interdependent within this proportional relationship, as together they constitute the total considered in this study. The structure of the switch in dominant relative contributions to coccolithophore CaCO$_3$ production is sharp, with respect to δ$^{13}$C$_{DIC}$ and SST environmental variability and noted with a red dashed line in (**a**) and (**b**); the structure of this switch is gradual with respect to the DIC/TA ratio, and the range of transitional values is noted with a shaded box in (**c**). See Methods for information about environmental data, corrections and calculations.

biogeographic distribution of these groups in both modern Atlantic Ocean and in sedimentary records of Holocene age (Supplementary Notes 3 and references therein). Also, the spatial relationship of each group with the preindustrial SST at latitudes points to a correspondence with the experimental determination of the SST ranges for optimal μ of those groups[74]. Building on the inverse relationship between cell size (coccolith size) and μ[35], together with the positive

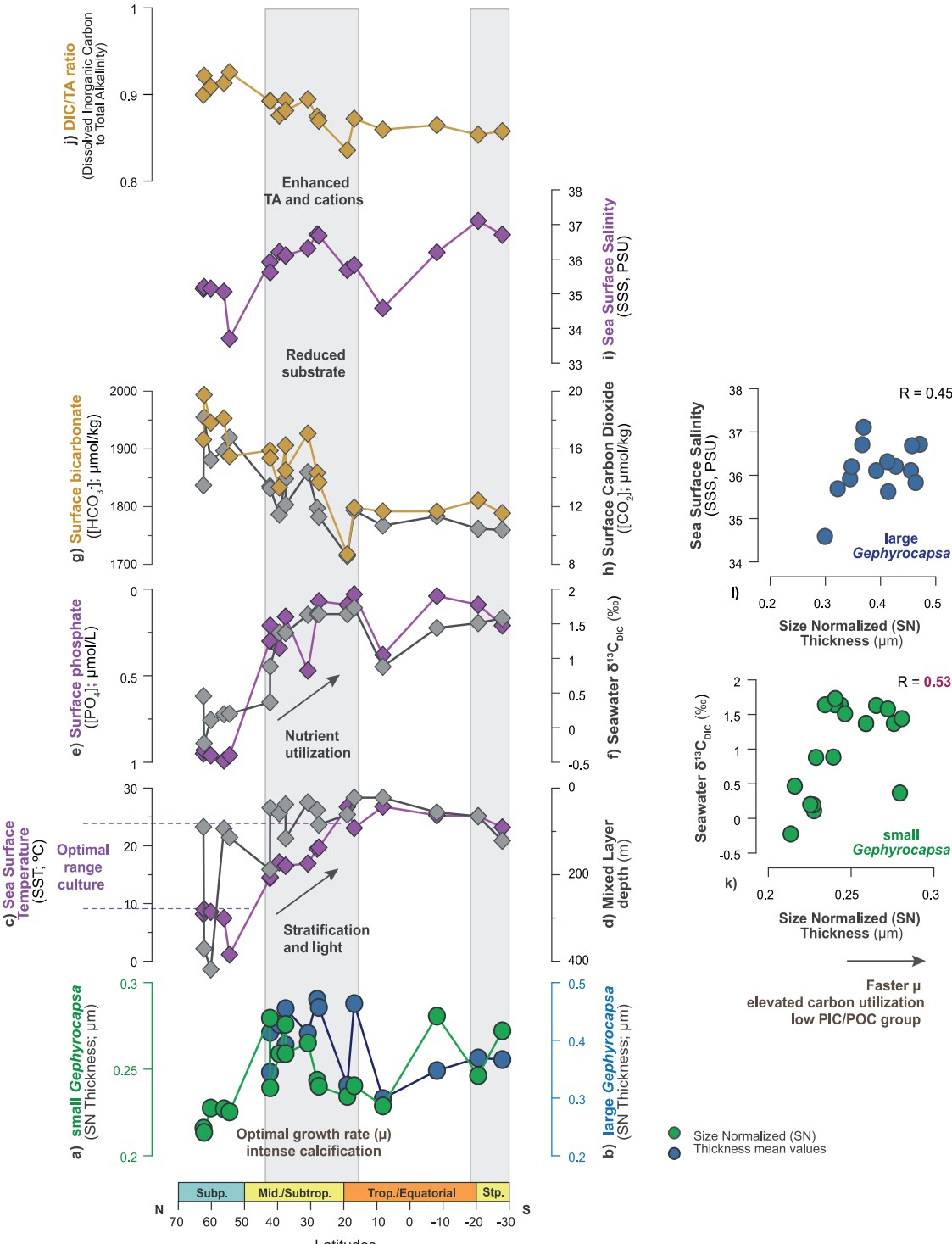

**Fig. 5 | Environmental controls on growth rate (μ) and calcification intensity in the low PIC/POC group. a, b** Calcification intensity profiles (SN Thickness; μm) for small *Gephyrocapsa* (green) and large *Gephyrocapsa* (blue), respectively. This parameter is considered equivalent to growth rate (μ) and carbon utilisation for this group. **c** Sea surface temperatures (SST; °C). The SST range associated with optimal μ in laboratory cultures[58,59] is indicated in the axis; **d** Mixed layer depth (m); **e** Surface phosphate concentration ([PO₄]; μmol/L); **f** Seawater carbon isotopic ratio of DIC (δ¹³C_DIC; ‰); **g** Surface bicarbonate concentration ([HCO₃⁻]; μmol/kg);

**h** Surface CO₂ concentration ([CO₂]; μmol/kg); **i** Sea surface salinity (SSS; PSU), representing qualitative variability in total alkalinity (TA) and cation concentrations (Supplementary Fig. 11); **j** Ratio of Dissolved Inorganic Carbon to Total Alkalinity (DIC/TA ratio). **k–l** Scatter plots showing the relationship between SN Thickness (μm), as an indicator of calcification intensity, μ and carbon utilisation in small and large *Gephyrocapsa*, respectively, against δ¹³C_DIC (‰) and SSS (PSU). See Methods for information about environmental data, corrections, and calculations.

correspondence between size in *C. pelagicus* and SST (Fig. 6l), this configuration indicates temperature-dependence for the distribution and optimal μ for these taxa. Likewise, a positive correspondence with surface [PO₄] and [CO₂] indicates optimal μ of this group under higher

availability, or saturation, in nutrients and carbon, consistent with the physicochemical characteristics of the environments north of ~40°N (Fig. 6m and n).

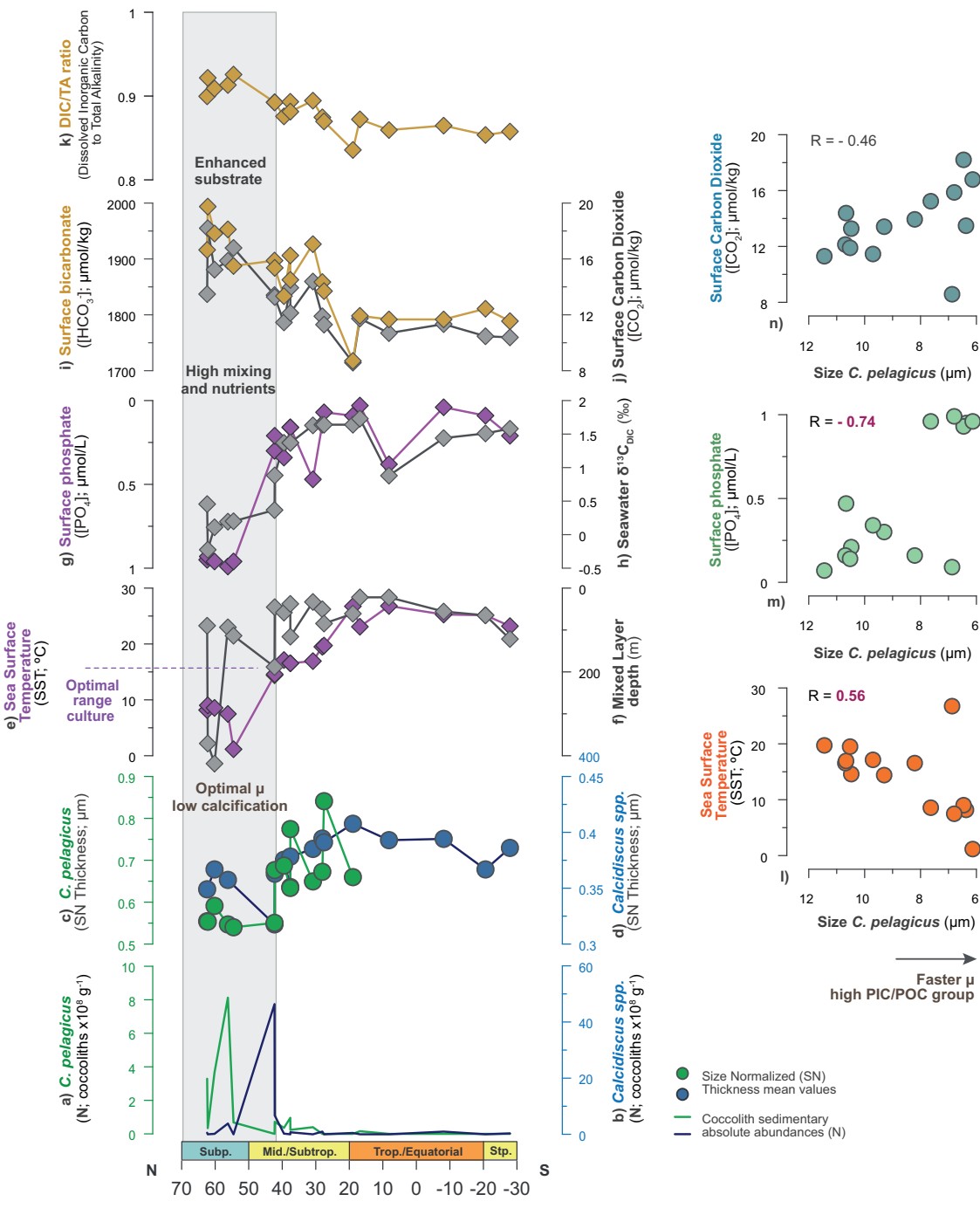

**Fig. 6 | Environmental controls on growth rate (μ) and calcification intensity in the high PIC/POC group. a, b** Qualitative variability in growth rate (μ) from coccolith absolute abundances (N; coccolith x $10^8$ $g^{-1}$) in *C. pelagicus* (green) and *Calcidiscus* spp. (blue) respectively; **c, d** Calcification intensity from SN Thickness (μm) in *C. pelagicus* (green) and *Calcidiscus* spp. (blue), respectively; **e** Sea surface temperature (SST; °C). The SST range associated with optimal μ in laboratory cultures[74] is indicated on the axis; **f** Mixed layer depth (m); **g** Surface phosphate concentration ([PO₄]; μmol/L); **h** Seawater carbon isotopic ratio of DIC (δ¹³C_DIC; ‰); **i** Surface bicarbonate concentration ([HCO₃]; μmol/kg); **j** Surface $CO_2$ concentration ([$CO_2$]; μmol/kg); **k** Ratio of Dissolved Inorganic Carbon to Total Alkalinity (DIC/TA ratio). **l–n** Scatter plots showing the relationship between μ of *C. pelagicus*, from its opposed relationship with coccolith size variability[35], with parameters (**k**) SST (°C; negative correspondence with μ), surface PO₄ (μmol/L; positive correspondence with μ), and surface [$CO_2$] (μmol/kg; positive correspondence with μ). See Methods for information about environmental data, corrections, and calculations.

## Implications for future and past coccolithophore calcite production

In the Atlantic Ocean, contrasting μ-calcification relationships between dominant coccolithophore groups indicate a gradient between reaction-limited calcification, in the higher latitudes, and transport-limited calcification in the lower latitudes. These two regimes determine the biogeographic distribution of the dominant high PIC/POC and low PIC/POC groups. A bimodal structure of their dominance on coccolithophore calcite contribution to sediments, with a boundary at ~40°N, is driven by the group-specific optimal μ within each group (Figs. 2 and 3). This gradient in their distribution and contribution to coccolithophore CaCO₃ likely reflects the ratio between metabolic rate

(cell demand) vs. carbon substrate (cell supply) modes in the low and high PIC/POC groups, respectively.

We propose that the position and environmental conditions at the boundary between the two regimes depend on the relationship between group-specific metabolic rate and carbon substrate availability. From the comparison with the mean annual environmental parameters in this study, the boundary between the two regimes is characterised by values of [PO$_4$] ~0.5 µmol/L, [CO$_2$] ~13 µmol/kg, DIC/TA ~ 0.9 and SST ~ 15 °C (Fig. 4 and Supplementary Fig. 10). These observations provide a useful indicator of conditions at the boundary for the modern communities, but should not be taken as absolute values for a calibration that can be applied universally. The intraspecific relationships between µ-calcification in groups and bimodal % CaCO$_3$ contribution relationships, along with the structure of the meridional gradient, may be persistent over time, and could be considered indicative of qualitative variability in those environmental parameters. Nonetheless, absolute values will likely vary across changing Cenozoic background conditions and through evolution within the calcifying phytoplankton lineages of both low and high PIC/POC groups.

In our record, toward the temperate to high SST environments in lower latitudes, CO$_2$ availability is reduced, due to low air-sea gas solubility (Figs. 5 and 6). In those environments, the prevalent low PIC/POC specimens, with high metabolic rates, are, therefore, most susceptible to mass transport limited precipitation and a more dendritic crystal growth (Fig. 2i, 5a, b and h and Supplementary Fig. 10). In the future, with increasing anthropogenic ocean acidification and warming, it is unclear whether the higher carbon availability (i.e., CO$_2$) in the surface ocean will be sufficient to meet a higher metabolic demand for calcification, due to elevated temperatures and µ of these cells. In the geological past, the progressive reduction in calcification of the specimens belonging to *Discoaster* spp. towards their eventual extinction in the glacial periods of the Pliocene to early Pleistocene[75,76] provides an anchor for this determination, suggesting that calcifying plankton are more sensitive to changing carbon availability, in the form of low CO$_2$, compared to the reduced temperature-dependent metabolic demand. We suggest CO$_2$ availability provided the dominant bottleneck in the past. Monitoring of the warmest areas of the oceans could reveal the fate of the low PIC/POC producers. The next emergent species (or subspecies) within this group could be of larger dimensions, if the increased carbon availability outpaces their escalating metabolic demand, or smaller and less calcified (or even uncalcified), if the temperature dependence of their metabolic demand outpaces the carbon supply.

Calcification intensities in low PIC/POC producers correspond with the µ of this group (Figs. 2e, f, 3a–c, f–h, k, m). Given the elevated metabolic activity of these specimens under conditions of relatively diminished cellular carbon supply, toward the mid-latitudes and subtropics, we suggest that increased allometry-normalised thickness change (i.e., SN Thickness, and also kse and morphometric PIC/POC) among low PIC/POC producers in well-preserved sedimentary records could serve as a proxy for elevated carbon utilisation in the past. Furthermore, the positive relationship between calcification intensity and maximum calcite production of the low PIC/POC group indicates the application of allometry-normalised thickness in coccoliths from this group as a qualitative indicator of enhanced calcite production in the past. As a geological example of this relationship, faster µ, intensified calcification, and a greater contribution to coccolithophore calcite production is similarly inferred for *Gephyrocapsa* specimens (low PIC/POC) of late Pleistocene age (mid-Brunhes, ~400,000 years ago), from morphometry and geochemistry[13,14]. In our study, the higher µ, calcification, and calcite production of the low PIC/POC group are associated with increased nutrient utilisation, reduced DIC/TA ratios, temperate SST, balanced light and a positive response to relatively elevated TA and cation concentration (Fig. 5c–f, i, j). These

factors may have contributed to the stimulation of *Gephyrocapsa* during the mid-Brunhes[13,14,77]. Such optimal nutrient and seawater chemical conditions during the mid-Brunhes episode could have arisen from intensified weathering and/or enhanced vertical mixing, provided that alkalinity resupply was sustained by high respiration rate-driven carbonate dissolution[12]. Inferences of modern environmental controls derived here, and associated with this geological interval, may also provide a useful reference for interpreting other geological episodes in which higher µ, calcification, and calcite production of the low PIC/POC group are identified.

The morphometric evidence related to high PIC/POC producers in this study shows that smaller coccoliths within this group are produced at the optimal µ at latitudes above ~40°N (Fig. 2 and Supplementary Fig. 5). The record of high PIC/POC specimens with smaller cell dimensions in the high latitudes may be a result of the faster µ[35]. We suggest this accelerated µ could drive the cellular carbon demand to outpace the extracellular carbon supply, leading to a reduced cellular quota of calcite (i.e., lowered calcification intensity) that has materialized through lower allometry-normalised coccolith thickness (i.e., SN Thickness). In other words, the high carbon requirements of this group may lead to calcification limitation under optimal µ, under the range of preindustrial to modern carbon availability conditions (Fig. 6a–d). The increased number of cells produced under optimal µ, nonetheless, trades off by driving a higher calcite contribution to sediments, despite the reduced coccolith quotas. The identification of coccolith size reductions in calcareous nannofossil sedimentary records containing representatives of the high PIC/POC group (i.e., mainly *C. pelagicus*), could be considered a qualitative indicator of faster µ and enhanced calcite production by this group. Elevated CO$_2$ supply in the seawater environment, together with low SST and vigorous mixing/nutrient advection, appear to be key environmental parameters controlling the optimal µ of the high PIC/POC species (Fig. 6). In the Holocene to modern Atlantic ocean, it is only in the higher latitudes where cellular carbon demand of more slowly dividing, but high carbon demanding cells (in comparison with low PIC/POC producers), can be met. We hypothesize that the identification of coupled faster µ (from reduced sizes and higher N) and allometry-normalised thickness increase (enhanced SN Thickness) in the geological past could indicate a greater availability of oceanic carbon substrate, or CO$_2$. Evolution of carbon availability through time may be revealed by finding this correspondence in nannofossil records. Exemplifying this, the reduction in prevalence and narrowing in the spatial distribution of the high PIC/POC specimens *C. pelagicus* across Cenozoic sedimentary records[78,79], suggest that the boundary between the high PIC/POC producers and the low PIC/POC community may have migrated progressively from the equatorial sectors towards higher latitudinal settings, with declining surface ocean carbon availability through the Cenozoic.

The boundary between the high PIC/POC and low PIC/POC contributors may, at first, appear to suggest very different calcite accumulation rates to the north versus the south. The calcite production rates comprise the number of cells and the amount of calcite per cell. From our sediments, the small *Gephyrocapsa* group (mostly constituted by *G. huxleyi*) generates 15 times more coccoliths than *C. pelagicus* (Fig. 2); *C. pelagicus* accumulates ~80 times more calcite than *G. huxleyi* (Fig. 2) and C. pelagicus divides at approximately one third the rate of *G. huxleyi*[46]. According to this rough estimation of the two endmembers of the population, the calcite production rates are of a similar order of magnitude across the boundary, despite the cellular-level differences in mass transport limitation.

## Methods
### Surface sediments
A set of nineteen surface sediment samples, or core-tops, collected from subpolar to equatorial regions of the Atlantic Ocean, were

selected for this study (Fig. 1 and Supplementary Fig. 1). Complete sample metadata information is available in the Supplementary Table 1. These samples represent relatively modern conditions, with maximum ages dating back to the late Holocene (i.e., < 3000 years). Detailed age information is provided in Supplementary Methods 1 and Supplementary Table 2.

All surface sediments in this study were collected from depths shallower than 3000 m, which are above the Atlantic Carbonate Compensation Depth (i.e., 3500 m) and the influence of Atlantic corrosive deep-water masses, such as the Northeastern Atlantic Bottom Waters, the Antarctic Bottom Water and the Circumpolar Deep Water (Supplementary Fig. 2). At the maximum depths of the studied locations, the saturation state of calcite ($\Omega_{Calcite\ depth}$) ranges from -1.4 to 2.2 (Supplementary Table 4). Since these values are well above the calcite undersaturation threshold at the chemical lysocline (i.e., $\Omega_{Calcite} = 1$)[80], a significant dissolution at the seabed can be ruled out. Further exercises to assess potential impacts of preservation over micropaleontological and morphometric data, including the comparison with independent datasets (i.e., sedimentology, computational productivity estimates, and well-preserved sedimentary/water column coccolithophore characterizations for reference) and statistical correlation analysis with variations in $\Omega_{Calcite\ depth}$ across sample depths, are included in the Supplementary Notes 1. Altogether, these support good preservation of the studied surface sediments.

The structure of sedimentary coccolith assemblages in samples fully aligns with the community composition and biogeographic distribution of natural oceanic communities and previously studied well-preserved surface sediments, whose mutual agreement proves their structure to be nearly invariant across the Holocene to modern periods (Supplementary Notes 3 and references therein). The % $CaCO_3$ in sediments remains highly stable across the studied locations (Supplementary Fig. 3) with very limited/slight deviations following the expected patterns of surface primary production, rather than diagenetic alteration (Supplementary Notes 1). Observations together support the representativeness of surface sediments as indicative of natural Atlantic conditions for the reconstruction of population dynamics, group-specific physiology (μ and calcification rates) and the assessment of calcite production patterns.

## Sample preparation and micropaleontological characterization

Microscope slides for analysing the structure of coccolith assemblages and morphometry from sediments were prepared using the settling technique proposed by Flores and Sierro[81]. This method ensures a homogeneous distribution of coccoliths, allowing an accurate specimen census quantification, the standardized comparisons between samples, and the calculation of absolute abundances, as coccoliths per gram of sediment (i.e., N; coccolith $g^{-1}$). The N values are calculated according to the formula outlined by Flores and Sierro[81]. The relative representation (%) of the different groups and species present in the sedimentary assemblages was also calculated.

Coccolith identification and census quantification were carried out using a ZEISS Axio Scope A1 microscope, equipped with cross-polarized light at a magnification of 1000x, at the Department of Earth Sciences of the University of Oxford (United Kingdom). Following the standard procedure for coccolith sedimentary analysis, a minimum of 300 specimens was counted per sample across a variable number of fields of view. This count ensures a statistically representative dataset, allowing for the reliable characterization of coccolithophore assemblages. After excluding preservation/dissolution effects, this data is considered representative of the coccolithophore community structure and composition, and its primary productivity variability, as standard practice (Supplemenary Notes 1 and 3).

For the identification of species and genus, the standard taxonomic criteria from Young et al.[82] and the guide of coccolithophore biodiversity and taxonomy Nannotax 3[83] were followed.

## Categorization of coccolithophore groups

The species belonging to the coccolithophore genus *Gephyrocapsa* spp. (including the small and large *Gephyrocapsa* taxa), *Helicosphaera* spp., *Calcidiscus* spp. and *C. pelagicus* are the targeted groups in this research. These taxa were selected on the basis of their ecological relevance, as dominant and widespread components of modern to Holocene Atlantic coccolithophore communities (Supplementary Notes 3 and references therein) and for the availability of standardized protocols for their morphometric characterization (Supplementary Methods 2 and 3).

Among the wide array of extant to Holocene *Gephyrocapsa* species, species-level identification is often challenging. As a result, ecological and physiological differences are generally addressed by distinguishing smaller specimens, *G. huxleyi* (i.e., the former *Emiliania huxleyi*), *G. parvula*, *G. ornata* and *G. ericsonii* from the larger specimens, *G. oceanica* and *G. muellerae*. Accounting for a separate characterization of these two ecological/physiological entities, the ~3/3.5 μm size threshold, common to sedimentary nannofossil research[84], is here applied for the differentiation between the small and large *Gephyrocapsa* taxa.

Based on their particulate inorganic carbon (PIC) to particulate organic carbon (POC) production ratios, coccolithophore taxa in this study are categorized into two functional groups: the low PIC/POC producers and the high PIC/POC producers. This classification follows a well-established framework from culture-based studies investigating coccolithophore physiology (see Chauhan and Rickaby[44] and references therein). According to those references, *Gephyrocapsa* taxa (small and large *Gephyrocapsa*) are categorized as a low PIC/POC group, generally exhibiting lower relative calcification rates and allocating a greater proportion of fixed carbon to organic matter. In contrast, taxa such as *Helicosphaera* spp., *Calcidiscus* spp. and *C. pelagicus* display comparatively higher calcification rates relative to organic carbon production and are, thus, classified as the high PIC/POC group.

## Image analysis and morphometric measurements

For image analysis, microscope slides were imaged using a Nikon DS-Fi1 8-bit colour digital camera coupled to a Nikon Eclipse LV100 POL microscope, equipped with circular polarization, at the Department of Geology of the University of Salamanca (Spain). The images were processed using the software C-calcita[85].

Following the standard protocol, the calibration of birefringence to calcite thickness was performed by measuring a cylindrically shaped *Rhabdosphaera* coccolith specimen[14,30,85,86]. For the analysis of each coccolithophore taxa, an average of 35 coccoliths was manually selected from pictures after calibration. The morphometric parameters of length (i.e., major and minor axis), volume, area, and mass of each of the individual coccoliths were automatically obtained with C-calcita[85]. Group-specific coccolith morphometries were further used to calculate cellular dimensions, thickness, calcification intensity indexes, and coccolithophore-related calcite concentration per gram of sediment, as indicated in the subsequent sections (see Supplementary Methods 2 and Supplementary Figs. 4–6). The dataset containing the complete set of individual coccolith morphometric measurements has been deposited in the Zenodo repository (https://doi.org/10.5281/zenodo.19323650)[87]. The secondary calculations for average morphometric values and calcification indexes are provided as Supplementary Data 1.

## Estimates of primary productivity and growth rates (μ)

Net primary productivity and growth rates (μ) are essential parameters for assessing the production dynamics of coccolithophores. Primary productivity refers to the total carbon fixed through photosynthesis, typically measured as milligrams of carbon per square meter per day (mg C $m^{-2}$ $d^{-1}$), for water column estimates. In contrast, μ indicates the

rate of biomass increase over time, commonly expressed as cellular divisions per day (div. day$^{-1}$; see Falkowski and Raven[88] and references therein).

In the absence of significant dilution or dissolution processes, micropaleontological proxies from sedimentary assemblages, such as N, serve as semi-quantitative indicators of changes in net coccolithophore primary productivity (Supplementary Notes 1 and 3). Absolute abundance values (N) and group-specific relative abundances (%) are included in the Supplementary Data 1. We explore the indication in Gonzalez-Lanchas et al.[39], that N may inherently provide qualitative information about the variability of coccolithophore μ by comparison with independent μ parametrizations. For the obtention of an independent profile of μ variability along environments, we apply the cellular size to μ regression for *Gephyrocapsa* (low PIC/POC group) by Zhang et al.[36]. This calculation is grounded in the well-established inverse relationship between these two variables, as consistently observed across a wide range of laboratory culture experiments by Aloisi[35]. To apply this calculation, the cellular sizes (radius) of *Gephyrocapsa* coccolithophores were derived from the values of coccolith sizes, following the dimensional relationship by Henderiks and Pagani[34]. For the high PIC/POC group (*Helicosphaera* spp., *Calcidiscus* spp. and *C. pelagicus* spp. specimens), where no species-specific equation exists, we assume the same opposed relationship between cell size and μ reported by Aloisi[35].

A complementary parametrization of net coccolithophore μ is conducted following the regressions by Krumhardt et al.[60], with the utilisation of surface environmental parameters [PO$_4$], [CO$_2$] and SST. Due to the independent consideration of [PO$_4$], [CO$_2$], and SST for these calculations, discrepancies between the output absolute values and the natural μ data are anticipated, as the unique effect of each parameter is unrealistic under natural conditions. Assessment of absolute values is not the primary focus of the study; rather, the emphasis lies on evaluating the consistency of latitudinal variability trends, along the primarily assessed μ, from N and coccolith/cell size variability, in this research. The different μ outputs are reported in the Supplementary Fig. 8 and included in the Supplementary Data 1.

## Coccolithophore calcification intensity and calcite production

Coccolithophore calcification intensity describes the amount of calcite produced per individual cell, serving as a measure of calcite investment at the level of the individual organism. Measurements of allometry-normalised thickness variability, the Size Normalized (SN) Thickness and elliptical shape factor (kse), are indicators of the changes in coccolith calcite storage. In the absence of dissolution, those indexes have been proposed to indicate changes in coccolithophore calcification intensity and have been widely used in sedimentary nannofossil studies (Supplementary Notes 1 and 2 and references therein).

The quantification of allometry-normalised thickness change and its correspondence to calcification intensity have been less commonly applied to taxa within the high PIC/POC group. Acknowledging the importance of developing more refined approaches for assessing calcification characterizations in these taxa, we evaluate the relationship between allometry-normalised thickness change in this group with relevant environmental and physiological parameters. A morphometric PIC/POC index is additionally calculated for the low PIC/POC group. Extended details about indexes calculation are provided in Supplementary Methods 2. Calcification indexes SN Thickness, kse, and PIC/POC are included in the Supplementary Data 1.

Coccolithophore calcite production reflects, for its part, the total amount of calcite output of the coccolithophore community, or a specific group, integrating both cell abundance and individual calcification. An approximation to coccolithophore calcite production is here assessed with the calculation of the concentration of coccolithophore-related calcite per gram of sediments, expressed as picograms of calcite per gram of sediment (pg CaCO$_3$ g$^{-1}$sediment).

This calculation considers the absolute abundance of coccoliths in sediments (N) together with average coccolith mass value for each group (i.e., small and large *Gephyrocapsa*, *Helicosphaera* spp., *Calcidiscus* spp. and *C. pelagicus*) at each studied sample. From these values, the relative contribution of the different groups to the coccolithophore calcite production, expressed as %, is calculated (see Supplementary Methods 3 and Supplementary Notes 4). These calculations are included in the Supplementary Data 1.

## Environmental parameters

A set of surface environmental data, including annual average sea surface temperatures (SST; °C), sea surface salinity (SSS; PSU), surface phosphate concentration ([PO$_4$]; μmol/L), total alkalinity (TA; μmol/kg) and pH was extracted for each studied location from GLODAPv2.2023[89–91] and World Ocean Atlas (WOA)[92–94]. All parameters were retrieved at a standardized depth of 100 m, using the software Ocean Data View (ODV)[95]. Annual maximum mixed-layer depths were also obtained from the same sources. The values of surface seawater dissolved inorganic carbon δ$^{13}$C (δ$^{13}$C $_{DIC}$) were obtained from the global preindustrial database corrected for the $^{13}$C-Suess effect by Eide et al[96]. Accurate data for each region, extracted at 200 m depth (the closest available to coccolithophore habitat depths) was selected over the WOA09 climatological annual mean gridded data, with the use of the Panoply Data Viewer (https://www.giss.nasa.gov/tools/panoply/). The annual SSTs were corrected to preindustrial values by subtracting the SST industrial anomaly at each region from Rayner et al.[97].

Because carbonate species concentrations are non-conservative and vary with speciation across latitudes[98] the individual concentrations of bicarbonate ([HCO$_3^-$]; μmol/kg) and dissolved CO$_2$ ([CO$_2$]; μmol/kg), the key substrates for coccolithophore photosynthesis and calcification, were calculated using the software CO2SYS[99]. In addition, the saturation state of calcite (Ω $_{Calcite\ depth}$) was estimated at the maximum depth of each studied location. A full compilation of all calculated and extracted parameters, that also includes the carbonate ion concentration ([CO$_3^{2-}$]; μmol/kg) and hydrogen ion concentration ([H$^+$]; nmol/kg), is provided in Supplementary Table 4.

## Potential biases and limitations

In our study, we highlight the mechanism driving a meridional gradient in coccolithophore calcite production, grounded on the contrasting μ-calcification relationships of dominant calcifiers. Although we point to semi-quantitative environmental relationships, we highlight that absolute calibration requires further work on comparison between timing and depth of production with appropriate seasonal and water column conditions.

The environmental conditions across the studied regions are dictated, to a first order, by the physical oceanographic variations, from a well-mixed water column at the higher latitudes to a more stratified water column in the lower latitudes. As a result, the majority of environmental parameters considered here (i.e., nutrients and carbonate system) parallel this physical oceanographic gradient. Although superimposed seasonal variance in each of the regimes should be acknowledged (i.e., mostly SST and light conditions), the structure of the chemical gradient across the latitudes remains robust. We note that coccolithophores prevail under higher light and lower nutrient conditions, which vary latitudinally with season, and, so, have opted for an approach based on mean annual averages, which can be applied across the entire transect for simplicity, and we reaffirm that sediments are a true integrator of all seasons and even multi-centennial change.

Compared to the meridional structure of the signal, variance over the maximal ~3,000 years represented by our sediments (i.e., ± 0.5/1 °C in SST[100]) remained overall low to minimal[101,102]. Holocene climate-related indices, including CH$_4$, CO$_2$, as well as δ$^{18}$O signals from the marine benthic stack, appear relatively stable over the last ~5,000

years[103], suggesting that minor stratigraphic inaccuracies, or differences, are not of great significance. We emphasise that surface sediments integrate surface ecological and environmental signals over timescales of a few thousand years, and their resolving power is limited to this time resolution. Modern to pre-industrial environmental conditions considered in this research provide an appropriate baseline for the comparison and qualitative assessment conducted here. Our approach develops an understanding of sedimentary parameters of physiology and coccolithophore calcification, which may be applied to the past.

## Statistical analysis

The Pearson correlation coefficient (R) and its level of significance (*p*-values) are calculated to assess the relationship between the proxy data generated and/or discussed in this study. The analysis was performed using the Statistica software. Correlations with *p*-values less than or equal to the threshold of 0.05 ($p \leq 0.05$) are considered statistically representative for interpretations in this study and marked in red bold in tables and figures.

## Data availability

The micropaleontological, morphometric data, and secondary calculations in this study are provided in the Supplementary Data file. The raw morphometric measurements generated and used to source the calculations have been deposited in the Zenodo database and are publicly available under reference: González-Lanchas, A., & Rickaby, R. (2026). Coccolith morphometric measurements, calcification indexes, and micropaleontological data from Atlantic surface sediments [Data set]. Zenodo. https://doi.org/10.5281/zenodo.19323650.

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

## Acknowledgements

This research used samples provided by the repositories GEOMAR (Germany) and the Godwin laboratory for paleoclimate research of the Department of Earth Sciences of the University of Cambridge (United Kingdom).

## Author contributions

A.G.L.: Conceptualization, formal analysis, investigation, methodology, visualization, and writing – original draft, review, and editing. R.E.M.R.: Conceptualization, funding acquisition, and writing – review editing. K.H.B., H.M.S., J.A.F., and M.A.F.: Data curation, resources, methodology, and validation.

## Funding

AGL and REMR disclose support for the research and publication of this work from the Natural Environment Research Council UK (NERC) project PUCCA (*Photosynthetic Underpinnings of Coccolithophore Calcification*), NE/V011049/1, and the European Research Council (ERC) under the European Union's Horizon 2020 research and innovation program project SCOOBi (*Seeking Constraints on Open Ocean Biocalcification*), 101019146. JAF discloses support for research from the project PICTURE funded by the Spanish Ministry of Science and Innovation, PID2021-128322NB-I00.

## Competing interests

The authors declare no competing interests.
