## [Transparent Peer Review file · Nature Communications]

Atlantic sediments reveal interacting environmental and physiological controls on coccolithophore calcite production

Corresponding Author: Dr Alba Gonzalez-Lanchas

Version 0:

Reviewer comments:

Reviewer #1

(Remarks to the Author)

I enjoyed reading the study of González-Lanchas et al., which provides some new insights into the biogeography of extant coccolithophores across the Atlantic Ocean through the detailed analysis of coccolith assemblages preserved in core-top sediments. Although the biogeographic patterns of coccolithophore species are well-established, from multiple plankton and previous core-top studies over decades, the current study nicely frames these patterns in terms of cellular PIC/POC ratios and provides something of a new perspective on two biogeographic domains (north/south of 40°N) based on cellular carbon demand versus supply (high/low PIC/POC groups respectively). The data is sound, well-presented and supports the findings of the study. The findings also have some implications for how to frame our understanding of the responses of these two, biogeographically distinct, groups to past and future changes in carbon substrate availability, and the trade-offs with metabolic rates.

My only question is about the breadth of appeal of the findings to the readership of Nature Communications. I don't have a strong view one way or the other and would leave to editorial judgement. A longer format (e.g. Biogeosciences?), with the integration of more of the context and underlying data from the supplementary information might actually gain more traction for these findings within the target research community.

On a minor point, I spent some time studying Figure 3. I think it over complicates the figure having %CaCO₃ for both high and low PIC/POC groups plotted, as these are just reciprocal (the two data sets are just reflections of each other in the vertical plane) – they could be expressed as the relative contributions of the two groups from 0 to 1. I'd also question if the axes should be switched – with the environmental drivers as the independent variable to which the relative carbonate contribution of the two groups respond. There is clearly an important signal in this data, but it's not immediately apparent in the current framing where the transitions (gradual with SST and DIC/TA) and thresholds (d13CDIC) are in these plots, and which are important and why.

Reviewer #2

(Remarks to the Author)

Using nanofossils as effective proxies for paleoceanographic has its limitations, largely due to the fact that they resist best efforts to perform culturing studies that underpin some other microfossil proxies (like foraminifera). Coccolithophoridae are important contribution to the total ocean inorganic carbon budget, and to global productivity in many parts of the world's present and past oceans. Thus, any successes at further ground-truthing the variations in morphology and calcification of coccos (their primary micropaleontologic attributes) are welcome.

This study attempts to provide a “modern” groundtruth of the cocco proxy by analyzing an array of coretop samples that span a critical productivity and alkalinity gradient in the Atlantic Ocean. The coretop samples span ocean geochemical gradients, including a critical one (the 40° N line) that represents significant changes in nutrients and alkalinity (lower phosphate, lower SST, higher alkalinity above 40°N). The study finds a gradient between reaction limited calcification in the higher latitudes and transport limited calcification in the lower latitudes, with distinctively different oceanographic conditions for high PIC/POC groups (above 40°N) compared to low PIC/POC groups (below 40°N). This careful analysis of cocco groups, their PIC/POC ratios, and their calcification yield a composite proxy for oceanographic conditions that can be used in

paleoceanographic reconstructions in a way that can substantially supplement other proxy signals.

The unique aspect of this study is that they have effectively opened up a bit of a black box when it comes to interpreting nanofossil records, substantially refining what the morphological, assemblage, and PIC/POC ratios mean in terms of ocean chemistry and productivity, beyond the simple high/low productivity interpretations of the past.

A limitation of this study is that it didn't attempt to apply this new, refined perspective to marine sediment cores to see how it holds up. I am more familiar with proxy calibration/improvement efforts that also test the fidelity of the proxy using paleo records of some kind. Nevertheless, the contributions here are unique, and may help to further expand the use of nanofossil paleo records.

Reviewer #3

(Remarks to the Author)

In "Environmental controls on coccolithophore calcite production in the Atlantic Ocean," González-Lanchas present an impressive micropaleontological dataset of sedimentary coccolithophores, with samples spanning from the subpolar North Atlantic to the subtropical South Atlantic. They use cell size to estimate growth rate, coccolith concentrations to estimate primary productivity, and size-normalized thickness to estimate calcification, which they shows a clear latitudinal gradient at ~40N, dividing the northern high PIC/POC communities from the southern low PIC/POC communities, which corresponds to several gradients in physical and biogeochemical properties.

As a oceanographer with a background studying present-day coccolithophores, I cannot comment much on the micropaleontological approach: while the method appears to be an innovative use of the sedimentary dataset, and shows very interesting relationships with other environmental data, I will defer to a reviewer with micropaleontological expertise to assess the robustness of the method. I will say, perhaps further demonstrating my ignorance in paleoceanography, that it was not clear how much environmental variability there was across the late-Holocene samples, and how temporal variability in calcifier deposition rates may influence how the data is interpreted. The environmental data are from GLODAP and WOA, with SST corrected to preindustrial values, but how similar were the temperatures that the fossils experienced to the preindustrial SSTs? Were the seasonalities/El Niño intensities similar? Have any of these environmental parameters been estimated from the sediment samples? How well sampled are each of these areas within the database (e.g. is there a sampling bias to a certain time of year)? The manuscript may reach a broader oceanographic audience if these considerations are outlined.

In all, I believe the manuscript would be better framed along the lines of "Coccolithophore calcite production in the Atlantic Ocean in the Holocene" to bring more focus on the interesting paleo methods applied, with the associated patterns with the environmental data for the discussion, which I recommend separating from the results section.

Version 1:

Reviewer comments:

Reviewer #1

(Remarks to the Author)

I'm happy with the changes made and would recommend for publication.

There are some minor points / typos to be corrected:

Line 69 – the 'production' of carbon? Would fixation be better here? Or production of particulate inorganic / organic carbon? They are not 'producing' carbon...

Line 120 – double 'the'

Line 401 – 'specimens' such as *G. oceanica* -> should be species such as...

Line 424 – 'appear to rely' -> "appear to be more responsive to cation availability"?

Line 444 why 'cooler/milder SSTs' – implies 'cooler than the coldest' (earlier part of the sentence) – isn't it just 'toward milder SSTs'?

Line 502 – "glacial periods of the Pliocene" -> glacial periods of the Pliocene to early Pleistocene. And consider more recent ref: <https://doi.org/10.1016/j.palaeo.2018.05.043>

Line 522 – "400.00" -> "400,000"?

Line 554 – don't need "increasingly" -> "the reduction in prevalence and narrowing..."

Figure 1 – DIC scale mislabelled

Reviewer #2

(Remarks to the Author)

This revision successfully addressed my minor concerns with the manuscript. As it is, it constitutes an important work of broad interest to the community.

Gabriel Filippelli

Reviewer #3

(Remarks to the Author)

I am satisfied with the reviewer responses to my comments and, should the other reviewers agree, believe the paper is suitable for publication.

Version 2:

Reviewer comments:

Reviewer #1

(Remarks to the Author)

Happy with the changes.

Response to Reviewers

Reviewer #1 (Remarks to the Author):

I enjoyed reading the study of González-Lanchas et al., which provides some new insights into the biogeography of extant coccolithophores across the Atlantic Ocean through the detailed analysis of coccolith assemblages preserved in core-top sediments. Although the biogeographic patterns of coccolithophore species are well-established, from multiple plankton and previous core-top studies over decades, the current study nicely frames these patterns in terms of cellular PIC/POC ratios and provides something of a new perspective on two biogeographic domains (north/south of 40oN) based on cellular carbon demand versus supply (high/low PIC/POC groups respectively). The data is sound, well-presented and supports the findings of the study. The findings also have some implications for how to frame our understanding of the responses of these two, biogeographically distinct, groups to past and future changes in carbon substrate availability, and the trade-offs with metabolic rates.

We appreciate the view and perception of our study by the reviewer and acknowledge their constructive comments. Providing additional perspective on the interconnection between environment and coccolithophore species specific group physiology, and their implication into Atlantic coccolithophore calcite production patterns, is the central aim in this research. Facilitating a route for using relatively classic and widely applied methods by the community, as micropaleontology and morphometrical analysis is our goal. We feel satisfied about the appreciation of the reviewer on those aspects. Following the subsequent recommendations in their report, the following changes have been conducted in the revised version of our manuscript and are presented below as a point-by-point response:

My only question is about the breadth of appeal of the findings to the readership of Nature Communications. I don't have a strong view one way or the other and would leave to editorial judgement. A longer format (e.g. Biogeosciences?), with the integration of more of the context and underlying data from the supplementary information might actually gain more traction for these findings within the target research community.

We agree with the reviewer that extended context from the underlying data and discussion included in the Supporting Information text could help to improve the accessibility of our manuscript to a general audience. Accordingly, this revised version incorporates further information from the Supporting Information text, that is now better framed, discussed and/or more clearly called to in the main text. Specific modifications conducted to highlight are:

- The new Figure 1. This includes the location of our samples over a map of the distribution of calculated total carbon in surface seawater. We consider this new Figure to contribute the contextualization of the materials in this study while providing key information about an environmental parameter that is lately discussed as an important control.
- Additional background information from the Supporting Information has been incorporated into the revised Main Text. These new elements are specifically located in the Results and Discussion sections (L 132–134; L 149–152; L 217–220 and L 250–252) as well as in the Methods section (L 590-599).
- The Supplementary materials have been revised with special attention to the Supplementary Notes 1 and 3; modifications conducted on those enable

facilitating the accessibility of the information included and their integration/coordination within the narrative in the main text.

On a minor point, I spent some time studying Figure 3. I think it over complicates the figure having %CaCO₃ for both high and low PIC/POC groups plotted, as these are just reciprocal (the two data sets are just reflections of each other in the vertical plane) – they could be expressed as the relative contributions of the two groups from 0 to 1. I'd also question if the axes should be switched – with the environmental drivers as the independent variable to which the relative carbonate contribution of the two groups respond. There is clearly an important signal in this data, but it's not immediately apparent in the current framing where the transitions (gradual with SST and DIC/TA) and thresholds (d13CDIC) are in these plots, and which are important and why.

A comprehensive revision of **Figure 3** (now revised Figure 4) has been carried out in response to the reviewers' comments. As indicated, the axes have been rearranged to present environmental drivers as the independent variables influencing the relative carbonate contributions. The plot now includes visual representation of the environmental boundaries and transitions, relevant for the discussion. To facilitate the readability and understanding of the figure (and mitigate the limitations pointed out by the reviewer) additional information about the natural two group mutual relationship has been clarified in the figure caption (L 268-275). To facilitate the evaluation of the group-specific data distribution patterns, we have opted to retain the representation of both groups as originally presented. Ultimately, extended information about the implications extracted from the observed environmental boundaries and thresholds has been now more explicitly discussed in the figure caption, and in the Results and Discussion section (L 346-354 and L 480-492).

We consider the revisions made satisfactorily address the reviewer's comments and indications and consider their constructive feedback has significantly improved the presentation of our manuscript. We remain available for any further clarification.

Dr. Alba Gonzalez-Lanchas, on behalf of the co-authors.

Reviewer #2 (Remarks to the Author):

Using nannofossils as effective proxies for paleoceanographic has its limitations, largely due to the fact that they resist best efforts to perform culturing studies that underpin some other microfossil proxies (like foraminifera). Cooccolithophoridae are important contribution to the total ocean inorganic carbon budget, and to global productivity in many parts of the world's present and past oceans. Thus, any successes at further ground-truthing the variations in morphology and calcification of coccos (their primary micropaleontologic attributes) are welcome. This study attempts to provide a "modern" groundtruth of the cocco proxy by analyzing an array of coretop samples that span a critical productivity and alkalinity gradient in the Atlantic Ocean. The coretop samples span ocean geochemical gradients, including a critical one (the 40° N line) that represents significant changes in nutrients and alkalinity (lower phosphate, lower SST, higher alkalinity above 40°N). The study finds a gradient between reaction limited calcification in the higher latitudes and transport limited calcification in the lower latitudes, with distinctively different oceanographic

*conditions for high PIC/POC groups (above 40*N) compared to low PIC/POC groups (below 40*N). This careful analysis of cocco groups, their PIC/POC ratios, and their calcification yield a composite proxy for oceanographic conditions that can be used in paleoceanographic reconstructions in a way that can substantially supplement other proxy signals.*

We thank the reviewer for their valuable comments and appreciation of our study. The motivation here is to decipher the varying interaction between dominant groups with the environment, providing additional ideas about the physiological-environmental dependence that dictates the patterns of calcite production. Utilising sedimentary coccolith/nannofossil records for this purpose enables, both, ensuring to register adaptive patterns beyond seasonal responses, while providing a powerful new route that could be further developed and applied to past sedimentary records. We are happy to see that the reviewer agrees, validates and supports this motivation and the contribution in this research. Following the subsequent recommendations in their report, the following changes have been conducted in the revised version of our manuscript and are below presented as point-by-point response:

The unique aspect of this study is that they have effectively opened up a bit of a black box when it comes to interpreting nannofossil records, substantially refining what the morphological, assemblage, and PIC/POC ratios mean in terms of ocean chemistry and productivity, beyond the simple high/low productivity interpretations of the past.

A limitation of this study is that it didn't attempt to apply this new, refined perspective to marine sediment cores to see how it holds up. I am more familiar with proxy calibration/improvement efforts that also test the fidelity of the proxy using paleo records of some kind. Nevertheless, the contributions here are unique, and may help to further expand the use of nannofossil paleo records.

While acknowledging that putting into practice the application of the refinement proposed in this research to marine sediment cores would positively complement our study, we consider that fully including said additional exercise in the present manuscript would substantially extend it, both in length and focus. Because of its novel nature, this study already required multiple layers of justification and discussion of the several datasets employed and methodological exercises conducted and included within the Supporting Information, to support the final stage in discussion about the physiological-environmental dependence. Efforts to qualitatively test the relationships observed and ideas proposed here to down-core studies will be produced in follow-up studies, which will provide the necessary space and context to explore specific temporal and regional scenarios in detail based on the foundations in this current manuscript. We reiterate, that we had already included an extensive last section in the manuscript: "*Summary and implications for future and past coccolithophore calcite production*" and some additional notes on this regard in other parts of the Results and Discussion section (L 182-191) that, indeed, place the implications of this research in the context of some key past geological intervals of changing coccolithophore production of calcite. Following the comments and perspective by the reviewer, we have, furthermore, revised this section in order to provide some additional views and more accurate determinations about the proposed routes for application of the relationships in our study and its future refinement/calibration (L 480-492).

We greatly appreciate the positive and constructive feedback from the reviewer; their insights helped us to clarify and strengthen some key aspects of our manuscript. We believe this revised version provides a more direct perspective on how the framework developed here could be extended and applied to the study of nanofossil sedimentary records.

Dr. Alba Gonzalez-Lanchas, on behalf of the co-authors.

Reviewer #3 (Remarks to the Author):

In “Environmental controls on coccolithophore calcite production in the Atlantic Ocean,” González-Lanchas present an impressive micropaleontological dataset of sedimentary coccolithophores, with samples spanning from the subpolar North Atlantic to the subtropical South Atlantic. They use cell size to estimate growth rate, coccolith concentrations to estimate primary productivity, and size-normalized thickness to estimate calcification, which they shows a clear latitudinal gradient at ~40N, dividing the northern high PIC/POC communities from the southern low PIC/POC communities, which corresponds to several gradients in physical and biogeochemical properties.

As a oceanographer with a background studying present-day coccolithophores, I cannot comment much on the micropaleontological approach: while the method appears to be an innovative use of the sedimentary dataset, and shows very interesting relationships with other environmental data, I will defer to a reviewer with micropaleontological expertise to assess the robustness of the method.

We thank the reviewer for their valuable comments and evaluation of our study. We consider their indications and perspective are highly valuable and followed them to further clarify and better integrate the considerations about the pointed elements in this revised version. The following changes have been conducted in the revised version of our manuscript and are below presented as point-by-point response:

I will say, perhaps further demonstrating my ignorance in paleoceanography, that it was not clear how much environmental variability there was across the late-Holocene samples, and how temporal variability in calcifier deposition rates may influence how the data is interpreted. The environmental data are from GLODAP and WOA, with SST corrected to preindustrial values, but how similar were the temperatures that the fossils experienced to the preindustrial SSTs?

The indications and comments from the reviewer about how to further and better frame the characteristics of the environmental dataset in this research are highly valuable for improving the presentation of our study. Considering the indications in this and the subsequent point, we have, on a first basis, implemented the information and discussion in our manuscript main text, specifically by including a new methods subsection “*Potential biases and limitations*” (L 740-767) and related arrangements in other parts of the text, revised Methods (L 590-599) and Results and Discussion (L 132-134; L 149-152; L 217-220; L 250-252; L 346-354; L 387-390 and L 480-492). To complement and ground this information, revisions have been made to Supplementary Notes 1 and 3 and their mention and integration within the main text. Beyond indicating this major modification on the regard of the reviewer’s indications we provide a detailed explanation about some key aspects and implementation.

Environmental variability across the age range between maximum sample age (i.e., late-Holocene) to preindustrial could be considered as negligible to the level of detail here: a range of thousand years (e.g., 3000 years) is typically within, if

not less, than the resolution of the majority of sedimentary records. Although some environmental conditions may have fluctuated over this timespan in some samples, those fluctuations are within the typical error at the resolution provided by the sediments and the methodology in this research (revised Methods, L 740-767). Together with this, the observed production patterns in our data are consistent with modern production conditions; and, critically, the patterns of distribution of the groups analysed in this research are fully consistent with both modern oceanic data and sedimentary/core top studies which, furthermore, evidence strong similarities in biogeographic dispersion and production trends (revised Supplementary Notes 1 and 3). Furthermore, Holocene climate-related indices such as CH₄ and CO₂ signals as well as δ¹⁸O signals from the marine benthic stack seem relatively stable during the last ~ 5000 years (e.g. Ruddiman et al., 2016, Rev. Geophys., 54, 93–118, doi:10.1002/2015RG000503) so that minor stratigraphic inaccuracies/differences are not of great significance. A comparison of (sub-recent) sediment trap data with surface sediments or sediment samples corresponding to the Late Holocene interval (0-2 ka) of very well-dated sediment cores also shows a good, albeit not perfect, correlation with the temperature data determined from planktic foraminifera (see Figure 4 in Makarova et al., 2025. Paleoceanogr. Paleoclimatol., 40, e2024PA004908, doi: 10.1029/2024PA004908). For coccoliths as well, a good correspondence exist between averaged annual coccolith flux and coccolith carbonate flux data from sediment traps and accumulation rates in underlying surface sediments (Sprengel et al., 2002, Deep-Sea Res. II, 49: 3577-3598; reference included in the main text).

We consider all this together reinforces that the assemblage distribution in our samples is insensitive to these small-scale variation over the age interval represented within our surface sediments samples and the suitability of considering the environmental modern/preindustrial parameter values as a reference for semi qualitative comparison in our study.

Were the seasonalities/El Niño intensities similar? Have any of these environmental parameters been estimated from the sediment samples? How well sampled are each of these areas within the database (e.g. is there a sampling bias to a certain time of year)? The manuscript may reach a broader oceanographic audience if these considerations are outlined.

This is an important observation by the reviewer. We have now implemented information on this regard in our revised manuscript, in order to address the different points raised and provide a more compelling view of the characteristics of the environmental data used for comparison. In sum, we want to emphasize that, due to the characteristics of sedimentary records already stated, the comparisons with annual averages of the parameters possibly more affected by seasonal change, provide the most robust foundation for the discussion presented in our work. This information has been clarified and discussed in the corresponding methodological description (L 590-599) and further reinforced in the new subsection included (L 740-767). Below we provide additional explanation:

The micropaleontological and morphometric data in this research, which form the basis for subsequent approaches for reconstruction of coccolithophore physiology are derived from surface sediments. The sedimentary record represents an integrated sum of variability, where seasonal precision is not discernible on the basis of this type of data. One possible approach could have used differing seasonal variability at different points across the latitudes for comparison between

sedimentary characteristics and season-specific growth months of coccolithophores. However, this would mean comparing signals from the spring/summer months in the higher latitudes with season-round signals in the lower latitudes. Resolving seasonal signals would be just possible using sediment traps, which is not the case of the approach used here, but something that could, indeed, constitute a potential route of refinement of the relationships proposed here in future research.

The majority of the environmental parameters assessed for comparison with the assemblages in our sediments are dictated by the physical oceanography of the water column i.e. mixing in the higher latitudes and stratification in the lower latitudes. This major structure is impervious to seasonal variance. For simplicity such that the whole dataset can be treated via one comparator, we reinforce that comparison with the mean annual average serves as the best anchor for our sediment–environment parameter comparisons.

We completely acknowledge, nonetheless, that the exact absolute values of the environmental datasets that define the boundary between the high PIC/POC producers and the low PIC/POC producers may flex somewhat through further work accounting for appropriate seasonal comparisons to be applied to the north and south of the boundary. And this is now further explored and explained in the revised version, Methods (L 740-767) and results and Discussion (L 387-390; L 480-492).

In all, I believe the manuscript would be better framed along the lines of “Coccolithophore calcite production in the Atlantic Ocean in the Holocene” to bring more focus on the interesting paleo methods applied, with the associated patterns with the environmental data for the discussion, which I recommend separating from the results section.

We appreciate this recommendation and view from the reviewer that greatly contributed to enhance the presentation of our manuscript. The title of our research has been modified to “Atlantic sediments reveal interacting environmental and physiological controls on coccolithophore calcite production”. We consider this new title brings more focus to the nature and characteristics of this study, and the potential of the methods proposed here (i.e., the application of sedimentary records as a route to characterize coccolithophore physiology) to be applied in future paleo studies.

We sincerely appreciate the thoughtful and constructive feedback and perspective from the reviewer. Their comments have been very helpful in clarifying and refining our manuscript. We are grateful for the valuable input received and remain available for any further questions or clarifications.

Dr. Alba Gonzalez-Lanchas, on behalf of the co-authors.

Response to Reviewers

We feel glad that our revision satisfied the indications of the three Reviewers. We appreciate the recommendation received and acknowledge again their contribution for the improvement of the presentation of our manuscript.

Dr. Alba Gonzalez-Lanchas, on behalf of the coauthors

Reviewer #1 (Remarks to the Author):

I'm happy with the changes made and would recommend for publication. There are some minor points / typos to be corrected:

Line 69 – the ‘production’ of carbon? Would fixation be better here? Or production of particulate inorganic / organic carbon? They are not 'producing' carbon... **modified**

Line 120 – double ‘the’ **corrected**

Line 401 – ‘specimens’ such as *G. oceanica* -> should be species such as... **modified**

Line 424 – ‘appear to rely’ -> “appear to be more responsive to cation availability”? **modified**

Line 444 why ‘cooler/milder SSTs’ – implies ‘cooler than the coldest’ (earlier part of the sentence) – isn’t it just ‘toward milder SSTs’? **modified**

Line 502 – “glacial periods of the Pliocene” -> glacial periods of the Pliocene to early Pleistocene. And consider more recent

ref: <https://doi.org/10.1016/j.palaeo.2018.05.043> **revised and reference included**

Line 522 – “400.00” -> “400,000”? **corrected**

Line 554 – don’t need “increasingly” -> “the reduction in prevalence and narrowing....” **modified**

Figure 1 – DIC scale mislabelled **corrected**

Reviewer #2 (Remarks to the Author):

This revision successfully addressed my minor concerns with the manuscript. As it is, it constitutes an important work of broad interest to the community.

Reviewer #3 (Remarks to the Author):

I am satisfied with the reviewer responses to my comments and, should the other reviewers agree, believe the paper is suitable for publication.

.